# Malaria and Helminthic Co-Infection during Pregnancy in Sub-Saharan Africa: A Systematic Review and Meta-Analysis

**DOI:** 10.3390/ijerph19095444

**Published:** 2022-04-29

**Authors:** Minyahil Tadesse Boltena, Ziad El-Khatib, Abraham Sahilemichael Kebede, Benedict Oppong Asamoah, Appiah Seth Christopher Yaw, Kassim Kamara, Phénix Constant Assogba, Andualem Tadesse Boltena, Hawult Taye Adane, Elifaged Hailemeskel, Mulatu Biru

**Affiliations:** 1Armauer Hansen Research Institute, Ministry of Health, Addis Ababa 1005, Ethiopia; hawultachew@gmail.com (H.T.A.); elifhabesha@gmail.com (E.H.); mulatu.biru@ahri.gov.et (M.B.); 2Department of Global Public Health, Karolinska Institutet, 17176 Stockholm, Sweden; 3World Health Programme, Université du Québec en Abitibi-Témiscamingue (UQAT), Rouyn-Noranda, QC J9X 5E4, Canada; 4School of Health and Sports Sciences, University of Brighton, Brighton BN2 4AT, UK; a.s.kebede@brighton.ac.uk; 5Social Medicine and Global Health, Department of Clinical Sciences, Lund University, 22184 Lund, Sweden; benedict_oppong.asamoah@med.lu.se (B.O.A.); tapaliau7@gmail.com (A.T.B.); 6Department of Sociology and Social Work, Kwame Nkrumah University of Science and Technology, Kumasi 101, Ghana; scyappiah@knust.edu.gh; 7Directorate of Health Security and Emergencies, Ministry of Health and Sanitation, Freetown 00232, Sierra Leone; kassim10915@gmail.com; 8Research Unit in Applied Microbiology and Pharmacology of Natural Substances, Polytechnic School of Abomey-Calavi, University of Abomey-Calavi, Abomey-Calavi 526, Benin; esseconstant.assogba@gmail.com; 9Department of Medical Microbiology, Radboud University Medical Center, 6525 GA Nijmegen, The Netherlands; 10Child and Family Health, Department of Health Sciences, Lund University, 22184 Lund, Sweden

**Keywords:** co-infection, comorbidity, helminthic infections, pregnancy malaria, sub-Saharan Africa

## Abstract

Malaria and helminthic co-infection during pregnancy causes fetomaternal haemorrhage and foetal growth retardation. This study determined the pooled burden of pregnancy malaria and helminthic co-infection in sub-Saharan Africa. CINAHL, EMBASE, Google Scholar, Scopus, PubMed, and Web of Science databases were used to retrieve data from the literature, without restricting language and publication year. The Joanna Briggs Institute’s critical appraisal tool for prevalence studies was used for quality assessment. STATA Version 14.0 was used to conduct the meta-analysis. The *I*^2^ statistics and Egger’s test were used to test heterogeneity and publication bias. The random-effects model was used to estimate the pooled prevalence at a 95% confidence interval (CI). The review protocol has been registered in PROSPERO, with the number CRD42019144812. In total, 24 studies (n = 14,087 participants) were identified in this study. The pooled analysis revealed that 20% of pregnant women were co-infected by malaria and helminths in sub-Saharan Africa. The pooled prevalence of malaria and helminths were 33% and 35%, respectively. The most prevalent helminths were *Hookworm* (48%), *Ascaris lumbricoides* (37%), and *Trichuris trichiura* (15%). Significantly higher malaria and helminthic co-infection during pregnancy were observed. Health systems in sub-Saharan Africa must implement home-grown innovative solutions to underpin context-specific policies for the early initiation of effective intermittent preventive therapy.

## 1. Introduction

Globally, approximately 1.5 billion cases of infection from malaria and helminths pose a significant risk of mortality and morbidity to the population at risk including pregnant women and the foetus [1,2]. Recently, a total of 12 million incidences of gestational malaria were reported out of 33 million pregnancies in sub-Saharan Africa (SSA) [2].

Ten countries in SSA—Burkina Faso, Cameroon, The Democratic Republic of the Congo, Ghana, Mali, Mozambique, Niger, Nigeria, Uganda, and The United Republic of Tanzania—that were hard hit by malaria endorsed the “High Burden to High Impact Approach (HBHI)” [3]; this sets out four response mechanisms to malaria elimination—namely, a political will to reduce death associated with malaria, strategic information to deliver impact, better guidance and policies, and a coordinated national malaria response strategy [4,5,6,7,8]. However, the already fragile healthcare delivery in SSA has faced the doubled burden of malaria and the novel coronavirus (nCoV-2) pandemic, which has stalled the hard-won gains in the fight against malaria [9,10,11].

The burden of helminthic infection during pregnancy in SSA ranges from 11% to 31% [12]. The most common helminths associated with unintended pregnancy outcomes in SSA include *Hookworm* (32%) [13], *Ascaris lumbricoides* (52%) [14], *Trichuris trichiura* (2.9%) [15], and *Schistosomiasis* (13%) [16]. Concurrent infection for more than one helminthic species during pregnancy shows negative health consequences on birth and maternal outcomes similar to malaria parasitaemia [17,18].

The World Health Organisation (WHO) 2030 road map aims to establish an efficient helminths control program specifically for women of reproductive age. Nevertheless, helminths continue to constitute major public health problems for pregnant women in SSA [19,20,21]. Co-infection from malaria and helminths is a major indicator of global health inequality, and failure to tackle this health disparity slows down the race to realising universal health coverage and attainment of the Sustainable Development Goal (SDG)-3 [22,23,24,25,26,27,28,29].

### 1.1. Clinical Implications of Concurrent Malaria and Helminthic Infection in Pregnancy

Malaria during pregnancy increases the risk of miscarriage and stillbirth by 3 to 4 times, compared with pregnant women with no clinically confirmed malaria [30]. Helminths cause alterations in immune response and physiological changes that affect fecundity, due to induced immunological states, with resultant adverse effects on conception and pregnancy [31]. Anemia during pregnancy is the most common adverse health outcome caused by *Ascaris lumbricoides* [32,33], and *Hookworm* [34]. In addition, *Schistosoma mansoni* is also associated with anemia and undernutrition during pregnancy [34], while *A. lumbricoides* is implicated with gallbladder perforation [35]. Pregnancy malaria co-infection with *A. lumbricoides* and *Hookworm* has been associated with increased odds of *P. falciparum* infection [36,37], and the pathophysiology of pregnant women simultaneously infected with *Plasmodium* species and helminths revealed negative pregnancy outcomes such as anemia, fetomaternal haemorrhage, antepartum stillbirth syndrome, and low birth weight [38,39,40]. Malaria and helminths co-infection causes elevated and unregulated inflammatory biomarkers such as C-reactive protein and serum level Hepcidin, which results in reduced iron absorption during pregnancy [41,42,43,44,45,46,47,48,49,50,51]. In addition, comorbidity of *Plasmodium falciparum* and helminthiasis elucidates the incidence of cervical cancer among pregnant women [52,53,54]. Moreover, during pregnancy, malaria co-infection with *A. lumbricoides* has been associated with an increased odds of *P. falciparum* infection [55,56], and malaria–Hookworm co-infection is associated with risks of increased *Plasmodium* parasitaemia [57,58].

Currently, evidence on the burden of intestinal helminths and malaria co-infection, the nature of their interaction, and their impact on pregnancy is not well established in endemic countries [59,60,61]. Most of the studies conducted in SSA emphasized the negative health outcomes of infection from malaria and helminths among pre- and schoolchildren [62,63,64,65,66], while very limited attention has been given to the dire impact of concurrent maternal gestational nematode and *Plasmodium* species infection [67,68,69,70]. Therefore, this systematic review and meta-analysis synthesised the available data on the burden of malaria and helminthic co-infections and their interaction among pregnant women living in SSA. It will further highlight evidence-informed planning and implementation for the comprehensive elimination of co-endemic malaria and helminthic infections during pregnancy in SSA [71,72]. 

### 1.2. Operational Definitions 

**Malaria in pregnancy:** This is an adverse clinical condition developed by pregnant women after being infected by Plasmodium species, which increases the risk of anemia, stillbirth, spontaneous abortion, low birth weight, and neonatal death [73]. Infants born to mothers living in endemic areas are vulnerable to malaria from approximately 3 months of age, which is when immunity acquired from the mother starts to wane [74,75,76,77].

**Co-Infection:** This is a clinical condition of particular human health importance caused by the simultaneous infection of a host (human being) by multiple pathogen species, for instance, multiple parasite infections [78,79,80,81,82].

**Helminths:** These are worms that infect the gastrointestinal tract of humans upon accidental ingestion of their infective eggs [83].

## 2. Materials and Methods

### 2.1. Reporting

The Preferred Reporting Items for Systematic Review and Meta-analysis (PRISMA) statement guidelines were used to fully record and report the search results and the reasons for exclusion of studies [84] (Figure 1) (Appendix A). The review protocol has been registered in PROSPERO with registration code CRD42019144812 [85]. An updated guideline for reporting systematic reviews (PRISMA checklist 2020) was used to report the corresponding section of the manuscript with its detailed contents and items [86,87] (Appendix A). 

### 2.2. Search Strategy and Information Sources 

A robust search was performed on CINAHL, EMBASE, Google Scholar, Scopus, PubMed, and Web of Science databases to retrieve published and unpublished data from the literature. (Appendix A). No restrictions were made regarding the language and years of all publications. The Boolean operators “AND” and “OR” were used to combine the MeSH terms ““Hookworm Infections”[Mesh] OR “Ascaris”[Mesh] OR “Ascaris lumbricoides”[Mesh] OR “Ascariasis”[Mesh] OR “Trichuris”[Mesh] OR “Trichuriasis”[Mesh] OR “Schistosoma”[Mesh] OR “Schistosoma mansoni”[Mesh] OR “Schistosoma haematobium”[Mesh] OR “Schistosomiasis mansoni”[Mesh] OR “Schistosomiasis haematobia”[Mesh] OR “Intestinal helminthiasis” [Supplementary Concept] OR “Anemia”[Mesh] AND “Co-infection”[Mesh] OR “Comorbidity”[Mesh] OR “Malaria”[Mesh] OR “Malaria, Vivax”[Mesh] OR “Malaria, Falciparum”[Mesh] OR “Acute malaria” [Supplementary Concept] AND “Pregnancy”[Mesh] OR “Pregnant Women”[Mesh]” and text words “Hookworm Infections*[tw] OR Soil-transmitted helminthiasis OR Ascaris*[tw] OR Ascaris lumbricoides*[tw] OR Ascariasis*[tw] OR Trichuris*[tw] OR Trichuriasis*[tw] OR Schistosoma*[tw] OR Schistosoma mansoni*[tw] OR Schistosoma haematobium*[tw] OR Schistosomiasis mansoni*[tw] OR Schistosomiasis haematobia*[tw] OR Intestinal helminthiasis*[tw] OR Anemia*[tw] AND Co-infection*[tw] OR Comorbidity*[tw] OR Malaria*[tw] OR Malaria, Vivax*[tw] OR Plasmodium vivax*[tw] OR Malaria, Falciparum*[tw] OR Plasmodium falciparum*[tw] OR Acute malaria*[tw] AND Pregnancy*[tw] OR Pregnant Women*[tw]” to run key search topics. Potentially relevant studies were fully retrieved, including their citation details, and additional data were obtained from the reference lists of some of the articles selected for critical appraisal. 

### 2.3. Study Selection

All the identified citations were exported into the EndNote version 15.0 reference manager. Two independent reviewers (M.T.B. and E.H.) rigorously screened the titles, abstracts, and the full text of selected literature against the inclusion criteria. The double-check of the included studies was performed by a third reviewer (H.T.A.). Discussions were made among the reviewers to resolve disagreements that arose at each stage of the study selection process.

### 2.4. Eligibility Criteria 

**Inclusion Criteria:** Observational studies published in SSA, which reported the co-infection of malaria in pregnancy with helminths as their main outcome were eligible for inclusion. Studies published in all languages of SSA until 20 January 2022 were included.

**Exclusion Criteria:** Systematic reviews, studies with poor methodological quality after, and reports of studies conducted outside SSA were excluded. Studies that employed inappropriate sampling frames, inadequate sample sizes, and poor data analysis were excluded. Studies that reported malaria or helminthic infection alone during pregnancy were also excluded.

### 2.5. Quality Assessment

The Joanna Briggs Institute’s (JBI) standardised critical appraisal instrument for prevalence studies was used to assess the methodological quality of included studies [88]. The JBI checklist contains nine quality measurement items (Appendix A). Studies scoring 6 and above out of the 9 criteria were considered to have high quality to be included in the meta-analysis (Table 1). Two reviewers (M.T.B. and H.T.A.) independently screened the eligible studies, and a third reviewer (E.H.) was involved to resolve the disagreement. The observed risk of bias in this study is low (93%) (Table 1). Studies that employed appropriate way of sampling procedures, had a clear description of settings and target population, appropriateness and adequacy of subject recruitment, reliability, and validity of methods used for the identification of outcomes of interest that included no co-infected cases (numerator), and a clear description of the study population (denominator) were deemed quality articles for final meta-analysis (Table 2). 

### 2.6. Data Extraction

Data extraction was principally carried out by two reviewers (M.T.B. and E.H.). The validity and eligibility of the extracted data for the meta-analysis were cross-checked by a third reviewer (H.T.A.). Variables such as the name of the corresponding author and publication year, study design and data collection period, sample size and study setting, the test approaches for the diagnosis of malaria, and helminths were extracted (Table 2). In addition, data extraction tools were used to extract the percentage of infection from *Plasmodium falciparum*, *Hookworm*, *Ascaris lumbricoides*, *Trichuris trichiura*, Schistosomiasis, the burden of helminths, prevalence of malaria, and malaria–helminthic co-infections, respectively.

### 2.7. Outcome Measurement 

Malaria and helminthic co-infection during the gestation period were considered to occur when a laboratory-confirmed case of at least one Plasmodium and helminth species identified from blood and faecal bio-specimens was obtained from pregnant women [63].

### 2.8. Statistical Analysis 

A quantitative meta-analysis of eligible studies was performed to estimate the event rate (prevalence of malaria–helminthic co-infection during pregnancy) [89]. Based on the random distribution assumption, the prevalence of each disease condition was obtained from the individual study estimate (ES), which includes a standard error (seES) and lower and upper confidence intervals. The pooled estimates were calculated and reported with respect to the relative weight given for each study [90,91]. Egger’s regression test analyses were used to check the publication bias [92]. The standard chi-squared I2 test was used to test heterogeneity [93]. A random-effects model using the double arcsine transformation approach was applied [94]. Decisions made regarding the included studies were checked by sensitivity analyses test. Funnel plot asymmetry visual examination and Egger’s regression tests were used to check for publication bias [95]. The pooled magnitude of co-infection of pregnancy malaria and helminths in SSA were estimated by computing a forest plot with 95%CI. Microsoft Excel 2019 workbook was used for data collection. The meta-analysis was performed using STATA version 14.0.3.

## 3. Results

### 3.1. Literature Search

A total of 1525 publications (Figure 1) were obtained from PubMed, CINAHL, EMBASE, Google Scholar, Scopus, and Web of Science databases, after removing 167 duplicates (Appendix A). Following title and abstract screening, a total of 1367 articles were excluded. Furthermore, 27 studies were eligible for quality assessment, out of which 24 studies were included in the meta-analysis (Figure 1). 

### 3.2. Characteristics of Included Studies 

A total of 14,087 pregnant women from 24 eligible studies from SSA participated in this systematic review. Studies with the highest (n = 2,507) and lowest (n = 87) sample sizes were reported from Uganda and Nigeria, respectively (Table 2). Only six studies reported data on parity rate with primigravida (n = 1159) and multigravida (n = 1803). Ten studies were reported from Nigeria [96,97,98,99,100,101,102,103,104,105,106], and three studies were from Kenya [69,107,108] and Uganda [109,110,111]. Two studies were reported from Ethiopia [112,113], Gabon [114,115], and Ghana [116,117], respectively. The remaining studies were reported from Malawi [118] and Cameroon [119]. All of the studies included in the review were conducted using cross-sectional study designs (Table 2). The majority of the studies employed the Kato–Katz thick smear, followed by formalin-ether and MacMaster concentration techniques for the detection of helminthic infection from faecal specimens, while the conventional microscopic method was used for the detection of malaria parasites (Table 2). Funnel plot asymmetry visual examination indicated no publication bias (Figure 2).

### 3.3. Meta-Analysis

#### 3.3.1. The Burden of Malaria Infection

The prevalence of malaria ranges from 4.6% to 36.2% (Table 2). The lowest and the highest pooled prevalence of malaria were 15% (95%CI: 12%, 17%) and 42% (95%CI: 39%, 45%) (Figure 3). The overall pooled prevalence of malaria was 33% (95%CI: 25%, 41%) (Figure 3). 

#### 3.3.2. The Burden of Helminthic Infection 

The pooled prevalence of helminthiasis was 35% (95%CI: 25%, 45%) (Figure 4). The prevalence of *Hookworm* infection ranged from 2% to 69% (Table 2). The pooled prevalence of *Hookworm* infection was 48% (95%CI: 36%, 61%) (Figure 5). The lowest and the highest prevalence of infection from *Ascaris lumbricoides* were 2% and 75%, respectively (Table 2). The pooled prevalence of *Ascaris lumbricoides* were 37% (95%CI: 30%, 44%) (Figure 6). The prevalence of *Trichuriasis* ranged from 1% to 21.4% (Table 2). The pooled prevalence of *Trichuris trichiura* was 35% (95%CI: 25%, 45%) (Figure 7). Only six studies have descriptively reported the burden of Schistosoma mansoni with the lowest (1.3%) and highest (46.8%) levels (Table 2). 

#### 3.3.3. The Burden of Malaria and Helminthic Co-Infection 

The lowest and the highest prevalence rates of comorbidity with malaria and helminths were 3% and 69% (Table 2). The pooled prevalence of malaria and helminthic co-infection was 20% (95%CI: 15%, 26%) (Figure 8). 

## 4. Discussion

This study estimated the pooled prevalence of co-infection of malaria and helminths during pregnancy from a total of 24 eligible studies and 14,087 pregnant women in SSA. The pooled prevalence of comorbidity of malaria and helminths among pregnant women in SSA was 20%, ranging from 9% in Ethiopia to 40% in Ghana. The burden of simultaneous infection from Plasmodium and helminthic species among pregnant mothers living in Uganda and Kenya was similar (16%). This could be attributed to the poor implementation of the intermittent preventive treatment of malaria during pregnancy, barriers to access to clean water, and inadequate sanitation in these three countries [120,121,122,123]. To tackle the impact of malaria and helminthic comorbidity on pregnant mothers, the WHO Africa region must establish a malaria data-sharing hub that can serve as a shared evidence-informing centre [124]. This will be a game changer by enabling the health systems in SSA to allocate scarce resources by applying a combination of updated tools for intervention and elimination strategies [125,126,127].

The burden of malaria in the gestational period among women immune-compromised by helminthic infection in SSA was 33%. This finding was higher than those of studies in Colombia (3.4%) [128] and Ethiopia (12.72%) [129]. This implicates the challenges to global malaria elimination efforts and calls for a collective concerted effort from countries in SSA to implement context-specific and tailored, evidence-based malaria elimination interventions [128,129]. Pregnant women’s poor adherence to the use of prescribed prophylactic antimalarial drugs and preventive measures puts strain on the malaria elimination goal [130,131,132,133,134,135,136].

This implies a concerted need to intensify malaria vaccine coverage in SSA to save the lives of pregnant mothers, in addition to having preventive, therapeutic, and control strategies in place to end malaria during pregnancy [137,138,139,140,141,142,143,144,145,146,147,148,149,150,151,152,153,154,155,156,157,158,159,160,161,162,163,164,165]. Countries in SSA must make changes in their malaria elimination strategies by adopting context-specific, home-grown innovative solutions, learning from grassroots experience, and strengthening public-private partnerships [142,143,144,145,146,147,148,149,150,151].

Our review revealed that the pooled prevalence of helminthiasis among pregnant mothers in SSA whose immunity is weakened by malaria was 35%. Uganda had a burden of helminthic infection in pregnancy (70%), which was higher than Cameron and Malawi combined (22%). Hookworm (48%), *Ascaris lumbricoides* (37%), and *Trichuris trichiura* (15%), respectively, were the pooled estimates of the most prevalent helminths associated with unintended pregnancy complications in SSA. The findings of our study were higher than those reported as global burden of helminthic infection during pregnancy in terms of the aggregate (3.6%) and species-specific Hookworm (19%), *Ascaris lumbricoides* (17%), and *Trichuris trichiura* (11%) [152]. This could be attributed to the inadequate availability of water, sanitation, and hygiene services in SSA, which remains below the global target of 80 % [153,154,155,156]. The prevalence of *Schistosoma mansoni* and malaria was determined by narrative synthesis because only 6 studies from the eligible 24 articles were reported with (n = 692) pregnant women from five countries in SSA who were co-infected by malaria and *Schistosoma mansoni.* Only five countries in SSA have (n = 1159) and (n = 1803) pregnant women in primigravid and multigravida who were co-infected by malaria and helminths. 

### 4.1. Optimisation of Anti-Malarial and Anti-Helminthic Infections in Endemic Areas

Although there are universal malaria interventions such as bed nets and access to prompt diagnosis and treatment for pregnant women in malaria-endemic settings, universal access to sanitation and hygiene should be implemented to prevent malaria and helminths co-infection in women of reproductive age and schoolchildren in endemic settings [157]. Moreover, improved diagnostic tools are required to better quantify the burden of malaria–helminth co-infection, as this might help understand the burden of these infections for evidence-based planning and implementation of integrated control and elimination of both malaria and helminthic infections in co-endemic areas [158]. Future malaria vaccine development efforts might also need to understand the immune modulation in malaria–helminth co-infection for better consideration of the effect of the helminth–malaria infection in vaccine immunogenicity [159].

### 4.2. Ending Preventable Maternal Mortality due to Malarial and Helminthic Co-Infection

The global effort to end the preventable death of the mother caused by the comorbidity of *Plasmodium* parasitaemia and helminthiasis requires a concerted global health leadership and commitment [160,161]. Sustainable implementation of the water, sanitation, and hygiene (WASH) programs, combined with improving the practice of early initiation of effective intermittent preventive therapy, can avert unintended health consequences as a result of malaria in pregnancy [162,163,164,165,166,167]. Unavailability of a platform for sharing real-time data, poor financing, and inadequate political commitment, coupled with the lack of an enabling and empowering environment to use state-of-the-art technology for the development of anti-malarial and anti-helminthic vaccines in the clinical and biomedical research and innovations in SSA, continue to hinder efforts to bring context-based solutions to achieve SDG3 [168,169,170,171,172,173,174,175,176,177].

### 4.3. Implications for Practice, Policy, and Future Research and Innovation 

Ensuring adequate access and enforcing adherence to safety and hygiene practices among pregnant women and safeguarding gestational mothers from economically disadvantaged households by creating sustainable access to economic opportunities will be essential to meet the global effort to control, prevent, and eliminate helminthic infections in sub-Saharan Africa [178,179,180,181,182,183,184,185,186]. To meet the 2030 target of successful elimination of helminthic infection, health systems in SSA and their international development partners must enhance the capacity and uptake of promising vaccine technology and innovation to improve maternal outcomes following gestational treatment of intestinal nematodes to help guide clinical decision making [187,188,189,190,191,192,193,194,195,196]. Sustainable and inclusive financing must be in place for the cutting-edge research and prudent innovation to deeply investigate the clinical outcomes of immunogenicity of comorbidity of malaria and helminths among gestational mothers in SSA [197,198,199,200,201,202,203,204,205,206]. Given the presence of sub-patent asymptomatic malaria burden that cannot be detected by microscopy [207], and *P. falciparum* parasites with histidine-rich protein 2 (pfhrp2)and histidine-rich protein 3 (pfhrp3) gene deletions that can escape the current HRP2 based-RDTs detection [208], the estimated burden of malaria- helminth co-infection might be underestimated in these 24 articles. Therefore, future studies that investigate the public health impact of asymptomatic malaria in pregnant women living in helminth co-endemic settings should be undertaken for better policy decision making. 

## 5. Conclusions

Significantly higher levels of malaria and helminthic co-infection during pregnancy were observed. Existing interventions, such as deworming, prioritisation, and distribution of insecticide-treated bed nets and other control measures addressing pregnant women need to be highly encouraged. In addition, health systems strengthening gatekeepers and health policy framers in sub-Saharan Africa must implement home-grown, innovative solutions to underpin context-specific policies and practice for early initiation of effective intermittent preventive therapy for the prevention of malaria in pregnancy. Investments in reverse vaccinology to augment cutting-edge research and innovations in the comorbidity of gestational malaria and helminths through public–private partnerships must be implemented by sub-Saharan African countries and their international development partners. Tailored advocacy on focused antenatal care must be in place to inform and raise awareness among pregnant women regarding the health benefits of universal sanitation and hygiene coverage, together with the effective establishment of integrated community-level early detection and treatment of malaria and helminthic co-infection in sub-Saharan Africa.

## Figures and Tables

**Figure 1 ijerph-19-05444-f001:**
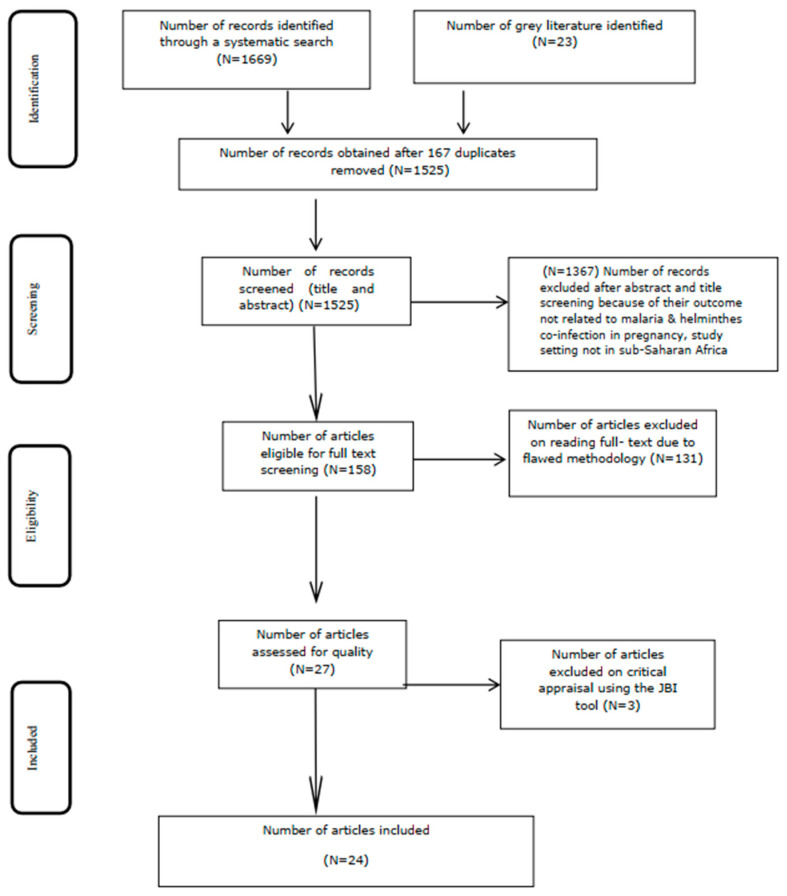
Flow diagram of the included studies. Moher, D. et al. Preferred Reporting Items for Systematic Reviews and Meta-Analyses: The PRISMA Statement. *PLoS Medicine*, **2009**, *6*(7).

**Figure 2 ijerph-19-05444-f002:**
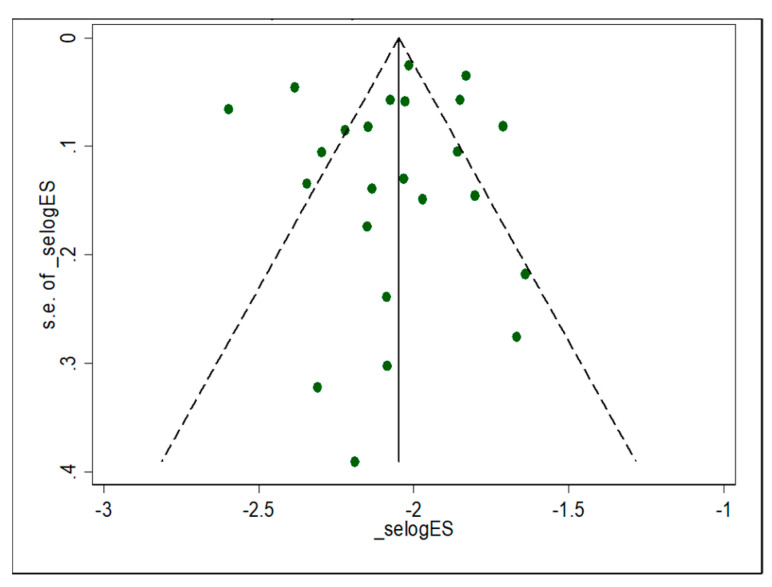
Funnel plot with pseudo 95% confidence limit of individual study estimates attributed with prevalence of malaria and helminthic co-infection among pregnant women in sub-Saharan Africa.

**Figure 3 ijerph-19-05444-f003:**
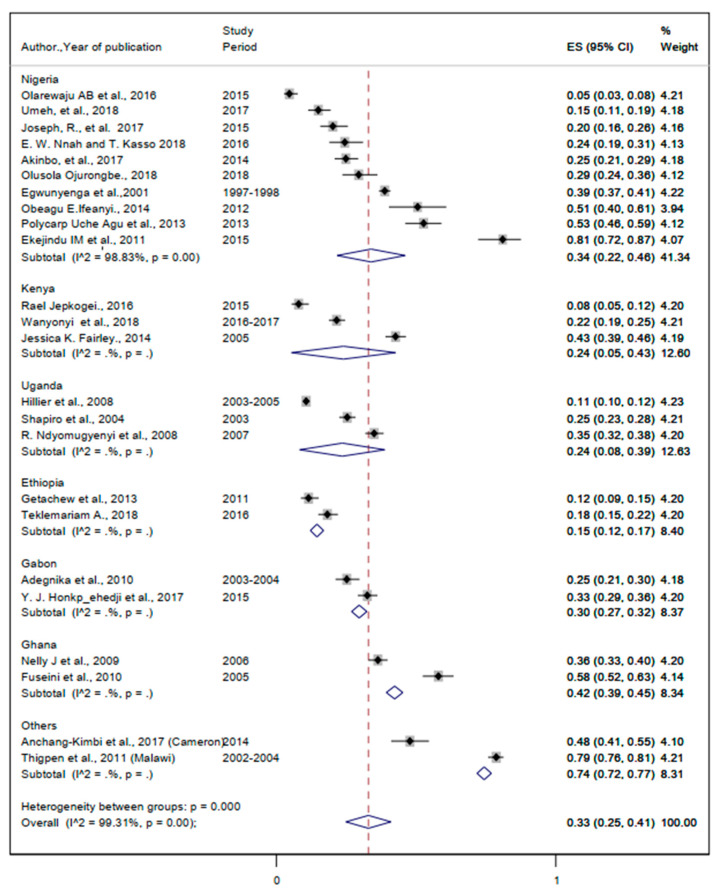
Forest plot for the overall and country—specific pooled prevalence of malaria among pregnant women in sub—Saharan Africa.

**Figure 4 ijerph-19-05444-f004:**
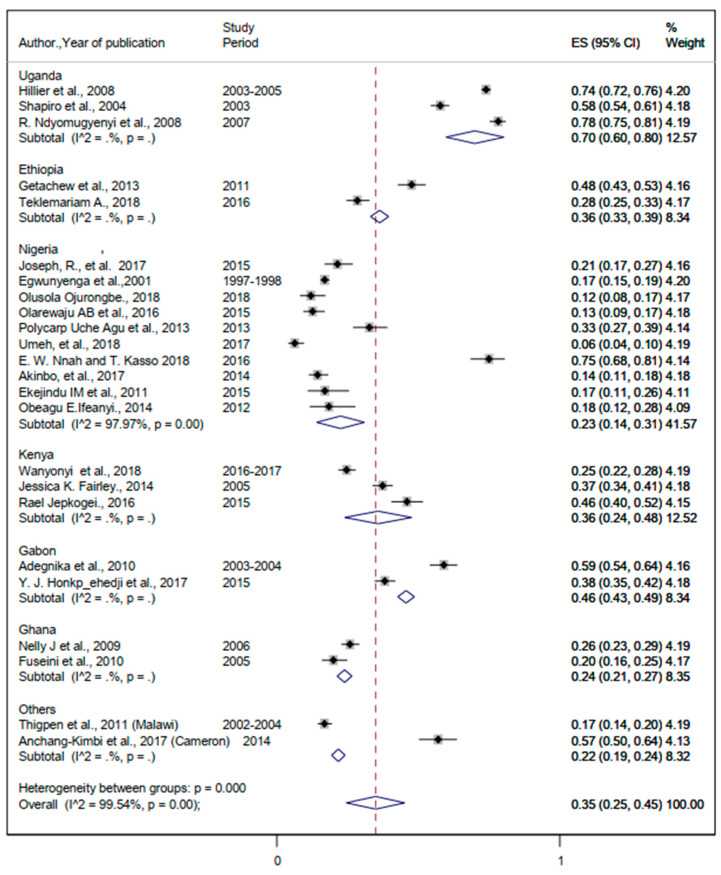
Forest plot for the overall and country-specific pooled prevalence of helminthic infection among pregnant women in sub-Saharan Africa.

**Figure 5 ijerph-19-05444-f005:**
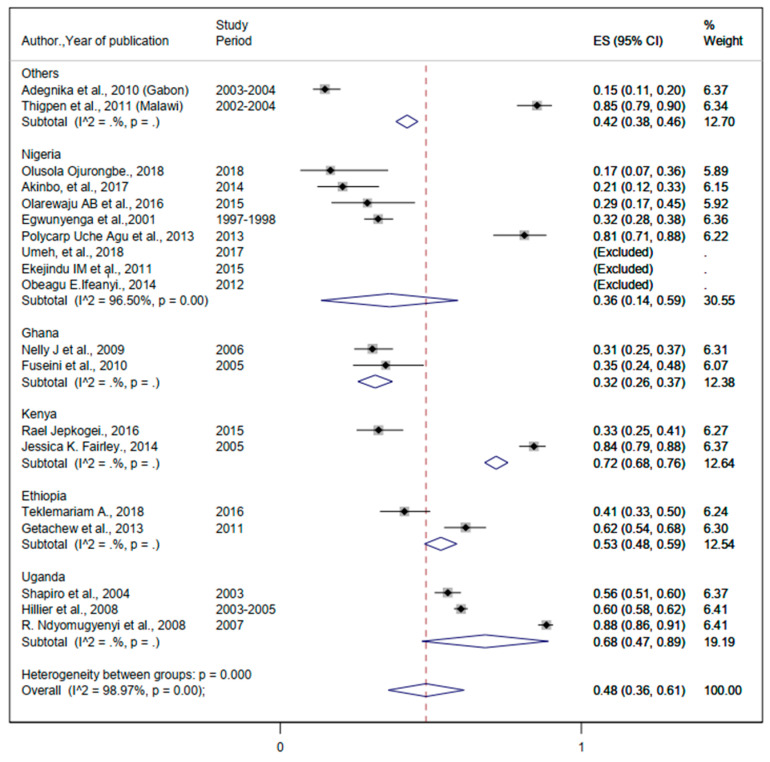
The proportion of Hookworm estimated from the overall helminthic infection among pregnant women in sub-Saharan Africa.

**Figure 6 ijerph-19-05444-f006:**
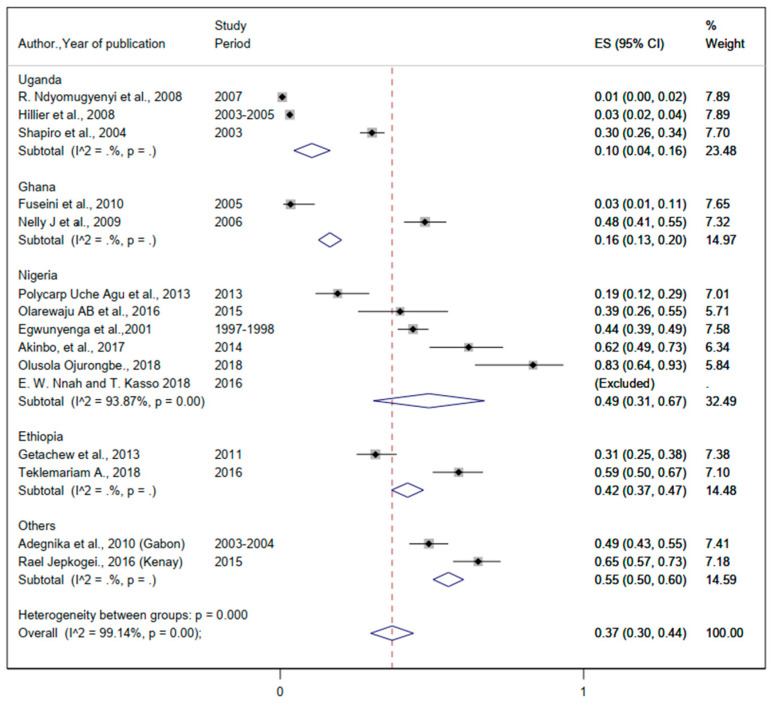
The proportion of *Ascaris lumbricoides* estimated from the overall helminthic infection among pregnant women in sub-Saharan Africa.

**Figure 7 ijerph-19-05444-f007:**
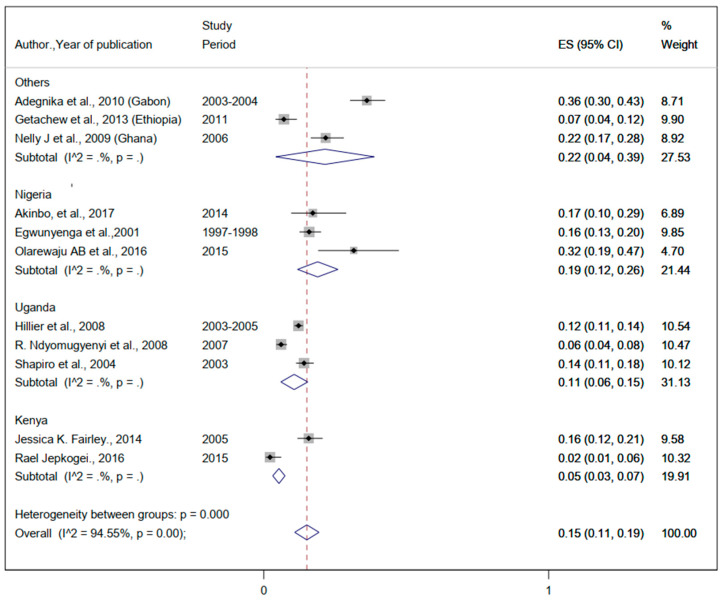
The proportion of *Trichuris trichiura* estimated from the overall helminthic infection among pregnant women in sub-Saharan Africa.

**Figure 8 ijerph-19-05444-f008:**
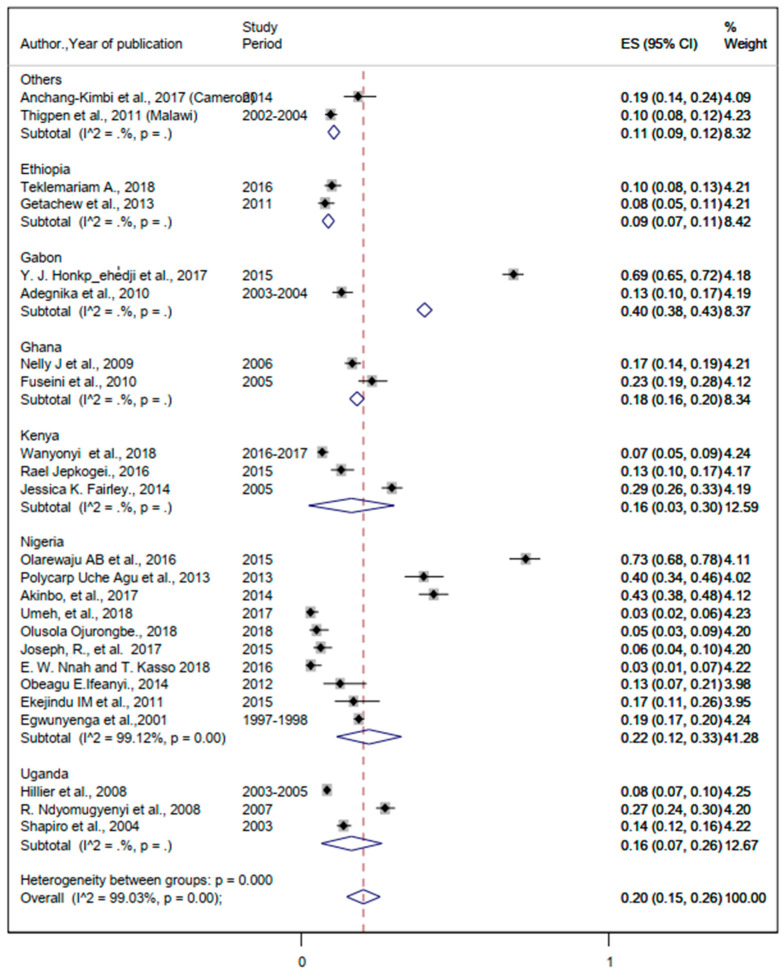
The overall pooled estimate and country-specific prevalence of malaria and helminthic co-infection among pregnant women in sub-Saharan Africa.

**Table 1 ijerph-19-05444-t001:** Quality assessment of the eligible studies.

Included Studies for Meta-Analysis		Study Level Bias Score	
S. No	Author, Publication year	Total No. Yes (Y)	Percentage of Yes (Y)
1	Hillier et al., 2008	9	100.00%
2	Getachew et al., 2013	8	89.00%
3	Joseph et al., 2017	9	100.00%
4	Wanyonyi et al., 2018	8	89.00%
5	Teklemariam A., 2018	8	89.00%
6	Egwunyenga et al., 2001	8	89.00%
7	Adegnika et al., 2010	9	100.00%
8	Nelly et al., 2009	9	100.00%
9	Shapiro et al., 2004	9	100.00%
10	Thigpen et al., 2011	9	100.00%
11	Olusola Ojurongbe	8	89.00%
12	Olarewaju et al., 2016	9	100.00%
13	Polycarp Uche Agu et al., 2013	9	100.00%
14	Ndyomugyenyi et al., 2008	8	89.00%
15	Anchang-Kimbi et al., 2017	8	89.00%
16	Umeh et al., 2018	8	89.00%
17	Nnah and Kasso, 2018	8	89.00%
18	Akinbo et al., 2017	7	78.00%
19	Ekejindu et al., 2011	9	100.00%
20	Ifeanyi., 2014	9	100.00%
21	Fairley, 2014	9	100.00%
22	Fuseini et al., 2010	7	78.00%
23	Masai, Rael Jepkogei, 2016	8	89.00%
24	Honkpehedji et al., 2017	8	89.00%
	Average bias score (%Yes)		93.00%

Subtotal Yes (Y) 93%. Subtotal No (N) 6.5%. Subtotal Unclear (U) 0%. Overall risk of bias assessment score was 93%. Remark: The risk of bias for each eligible study was calculated from the domain of nice criteria.

**Table 2 ijerph-19-05444-t002:** Descriptive summary of the eligible studies.

S. No	Author, Year of Publication	Year Study Conducted	Country	Study Design	Sample Size	Trimester			Parity		Test Approach for Malaria Diagnosis	Test Approach for Helminthiases	Prevalence of *Pf* Infection	Prevalence of *Pv* Infection	Prevalence of Any Malaria Infection	Prevalence of Malaria Associated Anemia	Overall Prevalence of Helminthiasis	Overall Prevalence of Malaria-Helminthiases Co-infection	*Hookworm*	*Ascaris* *lumbricoids*	*Trichuris* *trichuria*	*Shistosoma mansoni*
						1st	2nd	3rd	Primigravida	Multigravida												
1	Hillier et al., 2008	2003–2005	Uganda	Cross-sectional	2507						Microscopy	Kato-Katz thick smear	268 (11%)		268 (11%)		1693 (68%)		1112 (45%)	58 (2%)	226 (9%)	458 (18%)
2	Getachew et al., 2013	2011	Ethiopia	Cross-sectional	388	156	167	95	133	285	Microscopy	McMaster concentration technique			45 (11.6%)		159 (41%)	30 (7.7%)	114 (29%)	58 (15%)	13 (3.4%)	
3	Joseph, R. et al., 2017	2015	Nigeria	Cross-sectional	252				63	169	Microscopy	Formalin-ether concentration techniques+ wet mount			51 (20.2%)		54 (21.4%)	16 (6.3%)				
4	Wanyonyi et al., 2018	2016–2017	Kenya	Cross-sectional	750						Microscopy	Kato-Katz thick smear			21.60%	367 (48.9%)	24.70%	6.8%				
5	Teklemariam A., 2018	2016	Ethiopia	Cross-sectional	460						Microscopy	Formalin-ether concentration techniques	27 (5.9%)	55 (12%)	84 (18.3%)		198 (43%)	46 (10%)	54 (11.7%)	77 (16.7%)		
6	Egwunyenga et al.,2001	1997–1998	Nigeria	Cross-sectional	2104						Microscopy	Formalin-ether concentration techniques	762 (36.2%)		816 (38.8%)			394 (48.3%)	116 (5.5%)	156 (7.4%)	57 (2.7%)	28 (1.3%)
7	Adegnika et al., 2010	2003–2004	Gabon	Cross-sectional	388				111	277	Microscopy	Kato-Katz thick smear	98 (25%)				216 (64%)	15%	34 (8.8%)	112 (28.9%)	83 (21.4%)	
8	Nelly J et al., 2009	2006	Ghana	Cross-sectional	746	390	324	26	255	521	Malaria Antigen CELISA assay	Kato-Katz thick smear	271 (36.3%)		36.30%		192 (25.7%)	124 (16.6%)	59 (7.5%)	92 (12.3%)	42 (5.6%)	
9	Shapiro et al., 2004	2003	Uganda	Cross-sectional	856						Microscopy	Kato-Katz thick smear	217 (49.9%		217 (49.9%)		405 (47.3%)	118 (54.8%)	275 (32.1%)	149 (17.4%)	70 (8.1%)	
10	Thigpen et al., 2011	2002–2004	Malawi	Cross-sectional	848				412	436	Microscopy	Kato-katz thick smear	667 (37.6%)		667 (37.6%)	691 (81.5%)	143 (16.8%)	81 (9.7%)	122 (14.4%)			21 (2.5%)
11	Olusola Ojurongbe	2018	Nigeria	Cross-sectional	200	90	178	25			Microscopy	Formalin-ether concentration techniques	29.5% (59/200)				12% (24/200)	5% (10/200)	2.0% (4/200)	10.0% (20/200)		
12	Olarewaju AB et al., 2016	2015	Nigeria	Cross-sectional	300	32	116	152	185	115	Microscopy	Kato-Katz techniques	14 (4.6)		12 (4.0)			73.1% (219)	11 (3.6)	15 (5.0)	12 (4.0)	
13	Polycarp Uche Agu et al., 2013	2013	Nigeria	Cross-sectional	226	65	113	47			Microscopy	Kato-Katz techniques	119					90 (40%)	60 (26.5%)	14 (6.2%)		
14	R. Ndyomugyenyi et al., 2008	2007	Uganda	Cross-sectional	802						Microscopy	Kato-Katz techniques	281 (35%)					219 (16%)	554 (69%)	4 (0.5%)	38 (4.74%)	31 (3.87%)
15	Judith K. Anchang-Kimbi et al., 2017	2014	Cameroon	Cross-sectional	205	10 (4%)	125 (50%)	115 (46%)			Microscopy	Kato-Katz techniques	98 (39.2%)					38 (15.2%)				117 (46.8%)
16	Umeh et al., 2018	2017	Nigeria	Cross-sectional	300						Microscopy	Kato-Katz techniques	45 (15.0%)					9 (3%)	19 (6.3%)			
17	E. W. Nnah and T. Kasso 2018	2016	Nigeria	Cross-sectional	192						Microscopy	Kato-Katz techniques	47 (24.5%)			32 (16.7%)	1 (0.5%)	6 (3.1%)		144 (75%)		
18	Akinbo et al., 2017	2014	Nigeria	Cross-sectional	402						Microscopy	Kato-Katz techniques	100 (24.9%)				73 (18.2%)	173 (43.14%)	12 (3%)	36 (9%)	10 (2.5%)	
19	Ekejindu IM et al., 2011	2015	Nigeria	Cross-sectional	100						Microscopy	Kato-Katz techniques	81 (81%)					17 (13%)	17 (17%)			
20	Obeagu E. Ifeanyi., 2014	2012	Nigeria	Cross-sectional	87						Microscopy	Kato-Katz techniques	44 (51%)					11 (13%)	16 (18%)			
21	Jessica K. Fairley., 2014	2005	Kenya	Cross-sectional	696						Microscopy	Kato-Katz techniques	297 (42.7%)					205 (29.5%)	219 (31.5%)		41 (5.9%)	
22	Fuseini et al., 2010	2005	Ghana	Cross-sectional	300						Microscopy	Kato-Katz techniques	174 (58%)					69 (23%)	21 (7%)	2 (0.7%)		37 (12.3%)
23	Masai, Rael Jepkogei, 2016	2015	Kenya	Cross-sectional	300						Microscopy	Kato-Katz techniques	24 (8%)					39 (13%)	45 (15%)	90 (30%)	3 (1%)	
24	Y. J. Honkpehedji et al., 2017	2015	Gabon	Cross-sectional	678						Microscopy	Kato-Katz techniques	221 (33%)				259 (38%)	468 (69%)				

## Data Availability

The datasets during and/or analysed during the current study are available from the corresponding author on reasonable request.

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
