# Peer review of "Malaria and Helminthic Co-Infection during Pregnancy in Sub-Saharan Africa: A Systematic Review and Meta-Analysis"

_ijerph, 2022, doi:10.3390/ijerph19095444_

Round 1

Reviewer 1 Report

This study is interesting if we know that in Africa, there is a superposition of malaria and helminthiasis and that cases of co-infections are frequent, the fact of focusing on pregnant women makes this review even more relevant
However, a few points should be taken into account in the discussion
- The studies described in this review are based on microscopic diagnosis: the use of molecular tools which are more sensitive and would allow to highlight more infections. The authors should acknowledge that this could reduce the rate of co-infection reported in these studies
- the authors made a global analysis and did not try to see if the risk of co-infection was related to the age of the women or to their number of pregnancies. 
- Bilharzia is the second most common parasitic disease after malaria, how is it possible not to have cases of co-infection between plasmodium and S. mansoni?

-In the introduction, the authors cited a dozen countries with a high impact of malaria, can they highlight the impact of this malaria burden on the occurrence of co-infection with helminths
- In the practical aspects, the authors should propose a reflection on the optimization of anti-malaria and anti-helminthic interventions, to think about how to associate them in endemic areas.
- minor remarks:  
pge 284, no malaria vaccine in adults yet, and even less in pregnant women. In place, the IPT strategy should be reinforced

Author Response

We are very much grateful for the detailed comments and suggestions to our manuscript, which made it compelling and strong evidence synthesis. We have addressed all the concerns and questions raised. 

Reviewer 2 Report

A very interesting and well thought out review. Although the proposal is good and the search strategy is well designed, I have not been able to evaluate the tables, figures and annexes, as they are not integrated within the main text. Therefore, in my opinion, the review is incomplete. 

On the other hand, some comments that would improve the text would be:

  • Revise the main text, there are certain typographical errors (spaces, grammar, etc).
  • In the abstract, include the I2 index value.
  • Section 1.2. should be summarized, even moved to the discussion.
  • Section 1.3. These concepts should be integrated throughout the introduction.
  • What MeSH terms were included, how was the search equation, again, the figures and tables could not be reviewed.
  • Add the flow chart to see the search strategy for the articles at each step, as well as the figure of publication bias.
  • What was considered "poor data analysis" to be excluded? Why were studies with only malaria or helminthic infection excluded? (line 148-149).
  • The statistics section is very comprehensive.
  • Line 198. Why are 3 articles excluded to obtain the final N=24?
  • Section 3.3.2 should be rewritten to make it more understandable.
  • In the conclusions although I very much agree with them, they should indicate how their data contribute to the literature or society. Because the searches they did and the pooled data they obtain are important, since the indications they give are not part of their results.
  • The supplementary material is not necessary and in the funding "please add:" should be removed.
  • In the references: line 543 is not necessary and citation 125 is misplaced. 

Author Response

We are thankful for your earnest and thoughtful comments and questions you have raised to address in our manuscript, which we believe have made our manuscript compelling and strong evidence. We have addressed all the questions and included all the comments that we deem necessary. 

Round 2

Reviewer 2 Report

I thank the authors for this revised version of the article. In my opinion, it is understandable. However, I would like to suggest little modifications to make the article even more attractive:

  1. Table 1: the answers to each JBI question would not be necessary, if the two columns for the study level bias score appear.
  2. Figure 1 should be larger.
  3. Since it is a meta-analysis, the conclusions should contribute to society. 

Author Response

25 April 25, 2022

Dear Howie Huang, Section Managing Editor

We are very much grateful for the prompt response we have received to our manuscript. Please find our response to the key points raised by reviewer

Comments and Suggestions for Authors

I thank the authors for this revised version of the article. In my opinion, it is understandable. However, I would like to suggest little modifications to make the article even more attractive:

  1. Table 1: the answers to each JBI question would not be necessary, if the two columns for the study level bias score appear.

Response: We have addressed the reviewer’s comment accordingly. Please see the latest table 1 in the revised version of the manuscript attached.

  1. Figure 1 should be larger.

Response: Figure 1 has been made larger accordingly and uploaded as separate file. Please see the uploaded new updated Figure 1.  

  1. Since it is a meta-analysis, the conclusions should contribute to society. 

Response: We have addressed the reviewer’s comment in the conclusions section. Please see the track changed place in Section 5 (Conclusion section). “Tailored advocacy on focused antenatal care must be in place to inform and aware pregnant women regarding the health benefits of universal sanitation and hygiene coverage together with the effective establishment of integrated community-level early detection and treatment of malaria and helminthic co-infection in sub-Saharan Africa.”
